# The First Case of Congenital Myasthenic Syndrome Caused by a Large Homozygous Deletion in the C-Terminal Region of COLQ (Collagen Like Tail Subunit of Asymmetric Acetylcholinesterase) Protein

**DOI:** 10.3390/genes11121519

**Published:** 2020-12-18

**Authors:** Nicola Laforgia, Lucrezia De Cosmo, Orazio Palumbo, Carlotta Ranieri, Michela Sesta, Donatella Capodiferro, Antonino Pantaleo, Pierluigi Iapicca, Patrizia Lastella, Manuela Capozza, Federico Schettini, Nenad Bukvic, Rosanna Bagnulo, Nicoletta Resta

**Affiliations:** 1Section of Neonatology and Neonatal Intensive Care Unit, Department of Biomedical Science and Human Oncology (DIMO), University of Bari “Aldo Moro”, 70124 Bari, Italy; nicola.laforgia@uniba.it (N.L.); dlucrezia@yahoo.com (L.D.C.); dottcapodiferro@virgilio.it (D.C.); manuelacapozza26@gmail.com (M.C.); federico.schettini@uniba.it (F.S.); 2Division of Medical Genetics, Fondazione IRCCS Casa Sollievo della Sofferenza, 71013 San Giovanni Rotondo, Italy; o.palumbo@operapadrepio.it; 3Division of Medical Genetics, Department of Biomedical Sciences and Human Oncology (DIMO), University of Bari “Aldo Moro”, 70124 Bari, Italy; ranieri.carlotta@gmail.com (C.R.); antonino.pantaleo@uniba.it (A.P.); rosanna.bagnulo@uniba.it (R.B.); 4Neurology Unit, University Hospital Consortium Corporation Polyclinic of Bari, 70124 Bari, Italy; m_sesta@virgilio.it; 5SOPHiA GENETICS SA HQ, 1025 Saint-Sulpice, Switzerland; PIapicca@sophiagenetics.com; 6Rare Diseases Centre—Internal Medicine Unit “C. Frugoni”, Polyclinic of Bari, 70124 Bari, Italy; patrizia.lastella76@gmail.com; 7Medical Genetics Section, University Hospital Consortium Corporation Polyclinic of Bari, 70124 Bari, Italy; nenad.bukvic@policlinico.ba.it

**Keywords:** COLQ, congenital myasthenic syndrome, clinical exome sequencing, SNP-array

## Abstract

Congenital myasthenic syndromes (CMSs) are caused by mutations in genes that encode proteins involved in the organization, maintenance, function, or modification of the neuromuscular junction. Among these, the collagenic tail of endplate acetylcholinesterase protein (COLQ; MIM 603033) has a crucial role in anchoring the enzyme into the synaptic basal lamina. Here, we report on the first case of a patient with a homozygous deletion affecting the last exons of the *COLQ* gene in a CMS patient born to consanguineous parents of Pakistani origin. Electromyography (EMG), electroencephalography (EEG), clinical exome sequencing (CES), and single nucleotide polymorphism (SNP) array analyses were performed. The subject was born at term after an uneventful pregnancy and developed significant hypotonia and dystonia, clinical pseudoseizures, and recurring respiratory insufficiency with a need for mechanical ventilation. CES analysis of the patient revealed a homozygous deletion of the *COLQ* gene located on the 3p25.1 chromosome region. The SNP-array confirmed the presence of deletion that extended from exon 11 to the last exon 17 with a size of 19.5 Kb. Our results add new insights about the underlying pathogenetic mechanisms expanding the spectrum of causative *COLQ* mutations. It is relevant, considering the therapeutic implications, to apply suitable molecular approaches so that no type of mutation is missed: “each lost mutation means a baby treated improperly”.

## 1. Introduction

Congenital myasthenic syndromes (CMSs) are a group of genetically and clinically heterogeneous disorders with impaired neuromuscular transmission and symptoms such as fatigable skeletal muscle weakness generally confined to ocular, bulbar, or limb-girdle muscles [1]. CMSs can be inherited in either an autosomal recessive or autosomal dominant manner; however, the autosomal recessive manner is more frequently observed.

CMSs differ from myasthenia gravis, an autoimmune disorder that affects the same anatomic region, because of early age of onset, positive family history, rarity, and lack of response to immunomodulatory drugs. Clinical presentation of CMSs is highly heterogeneous and ranges from mild symptoms to severe manifestations, sometimes with life-threatening respiratory episodes especially in the first decade of life [2]. All subtypes of CMSs share the clinical features of fatigability and muscle weakness, but the age of onset, presenting symptoms, and response to treatment relate to different genetic defects [3]. CMSs should be suspected in the case of early-onset fatigable muscle weakness, which especially affects the ocular, bulbar, and limb muscles, positive family anamnesis, clinical and neurophysiological myasthenic findings with a negative antibody testing profile, and abnormalities of electromyography (EMG) [4]. Affected patients may also present dysmorphisms, neuropathic pain, seizures, pterygia, contractures, hyperlaxity of joints, abnormal speech, cognitive impairment, respiratory insufficiency, or skeletal deformities.

Onset can be prenatal, during infancy, childhood, or adolescence (rare). Clinical onset may sometimes be in adulthood with EMG changes not present in all muscles and intermittent [5].

The most common cause of CMS is a molecular defect impacting the function (CHRNA1, CHRNB, CHRND, CHRNE and CHRNG) or the clustering (RAPSN, DOK7, MUSK, LRP4) of muscle nicotinic acetylcholine receptors at the neuromuscular junction. CMS can also be caused by mutations in genes encoding proteins in the synaptic basement membrane (COLQ, AGRN, LAMB2), in the presynaptic release machinery (CHAT, VACHT, SLC5A7, SYT2, SNAP25B)**,** or in protein glycosylation (GFPT1, DPAGT1, ALG2 and ALG14) [3].

Intra- and inter-familial phenotypic heterogeneity have been described with the same genotype and a possible gender effect [6]. Thirty CMS disease genes have been identified, and all are functionally involved in the development and maintenance of neuromuscular endplate. Of these, the collagenic tail of endplate acetylcholinesterase (COLQ; MIM 603033) forms a triple helix collagenic tail required for the anchoring of acetylcholinesterase to the synaptic basement membrane. Different mutations have now been identified in each COLQ domain and, depending on their location, they can affect the assembly with catalytic subunits or prevent the formation of the triple collagen helix [3]. We present the first report of a homozygous deletion of the C-terminal region of the COLQ protein in a Pakistani boy with CMS and epilepsy born to healthy parents (first cousins) identified by the clinical exome sequencing (CES) approach integrated with a copy number analysis algorithm. This approach expands the genotype–phenotype correlation of CMS and improves genetic counseling and access to precision medicine.

## 2. Materials and Methods

### 2.1. Patient Recruitment

Written informed consent to perform genetic testing and further studies were obtained from the family using a form approved by the competent ethics committee in line with the principles of the Declaration of Helsinki and any other applicable local ethical and legal requirements (approval code 6631-prot. N° 93990/03/12/2020).

### 2.2. Clinical Exome Sequencing (CES)

Next generation sequencing analysis was performed on genomic DNA from peripheral venous blood (QIAamp DNA Blood Mini Kit) with a clinical exome sequencing panel kit. Approximately 11 Mb (114.405 exons) of the conserved coding regions that cover >4500 genes were enriched with >150,000 probes, which were designed based on human genome sequences (Sophia Genetics SA, Saint Sulpice, Switzerland). Library preparation and sequencing were performed according to the manufacturer’s protocol on MiSeq Instrument (Illumina, San Diego, CA, USA). The mean depth of coverage was 70×. Raw data were analyzed using SOPHiA™ DDM (Sophia Genetics SA) with algorithms for alignment including single nucleotide polymorphisms (SNPs), and insertions/deletions (Pepper™, Sophia Genetics SA patented algorithm), and copy number variations (Muskat™, Sophia Genetics SA patented algorithm). The raw reads were aligned to the human reference genome (GRCh37/hgl9), and an integrative genomics viewer (IGV) was used visualize the binary alignment map (BAM) files.

### 2.3. SNP-Array Analysis

High resolution SNP-array analysis of the proband and his parents was carried out by using the CytoScan HD array (Thermo Fisher Scientific, Waltham, MA, USA) as previously described [7,8].

This array contains more than 2.6 million markers for copy number variations (CNVs) analysis and approximately 750,000 SNP probes capable of genotyping with an accuracy greater than 99%.

Data analysis was performed using the Chromosome Analysis Suite Software version 4.1 (Thermo Fisher Scientific) following a standardized pipeline described in literature [8]. Base pair positions, information about genomic regions and genes affected by CNVs, and known associated disease have been derived from the University of California Santa Cruz (UCSC) Genome Browser, build GRCh37 (hg19).

## 3. Results

### 3.1. Clinical History

This was the fourth child of first cousin parents born at term (3470 g at 41 weeks) after an uneventful pregnancy. One neonatal death for unknown causes was reported. The child was discharged home after three days (Figure 1).

At 55 days of life (DOL), he was brought to the emergency department (ED) of a local hospital for dyspnea and cyanosis. He was intubated and ventilated and developed tonic-clonic fits during hospitalization that were treated successfully with phenobarbitone. He was then discharged after 15 days but was then admitted to our department at DOL84 because of hypotonia, weight loss, and difficult feeding. Weight, length, and head circumference were all <3° centile for his age. Marked hypotonia, dystonia, mild palpebral ptosis, and electroclinic fits (chaotic movements, hyperextension of arms and legs followed by marked hypotonia, apnea and cyanosis) with partial response to different drugs (levetiracetam, carbamazepine, midazolam, clonazepam, vitamin B6, and vigabatrin) were evident. Electroencephalography (EEG) showed theta–delta waves starting from the right occipital region with contralateral spread followed by the depression of the brain’s electrical activity linked to a cyanosis crisis (Figure 2A–D). Cerebral MRI revealed a normal brain structure except for mild enlargement of the subarachnoid space. He was discharged after four months at the age of eight months and was treated with levetiracetam with no new episodes of convulsions.

Palpebral ptosis became more evident at 12 months. The EMG revealed a decremental response to repetitive nerve stimulation on the deltoid muscle at a frequency of 3 Hz. The EMG showed a decrease in the amplitude of motor unit potential (MUP) with no post-increment. Treatment was started with noninvasive ventilation (NIV) during nocturnal sleep and pyridostigmine. Pyridostigmine was started at the dose of 0.78 mg/kg/day in four divided doses. There was only improvement of the ptosis, no effects on apnea and respiratory crisis. Pyridostigmine was then stopped at 18 months after genetic diagnosis of COLQ mutation, and both 3–4 diaminopyridine and salbutamol treatments were started with positive effects on muscular tone and a significant reduction in the apnea and respiratory crisis. The child is now 20 months old with a mild neuromotor delay. He can walk with support. NIV during sleep is still needed.

### 3.2. Genetic Findings

CES of the patient identified a homozygous microdeletion of part of the *COLQ* gene and heterozygous status in both parents (Figure 3A). The deleted exons identified by CES were exons 13–17. No other significant single nucleotide variants (SNVs) in genes related to the clinical features were detected. SNP-array analysis confirmed the homozygous deletion involving the 3p25.1 chromosome region. The deleted region was 19.5 Kb in size and was covered by 24 SNP-array probes. The SNP-array demonstrated the deletion of exons 11–17 of the COLQ gene, which was wider than that revealed by CES analysis. The discrepancy between the SNP-array and NGS results may reflect the limitation of the CNV-detection algorithm of the latter technology for large panels due to coverage fluctuations, as CNV resolution is based on the coverage levels of the target regions. No other CNVs were detected apart from known polymorphisms. The molecular karyotype of the patient according to the International System for Human Cytogenetic Nomenclature (ISCN 2016) is: arr[GrCh37] 3p25.1(15491478x1,15492150_15511615x0,15511740x1). The deleted region in 3p25.1 contains part of the *COLQ* gene (i.e., exons 11–17). Carrier testing in the parents was performed by chromosome microarray analysis (CMA) using the same platform (i.e., CytoScan HD Array) and resulted in heterozygous outcomes in both (Figure 3B).

## 4. Discussion

The usual clinical pattern of CMSs is characterized by abnormal fatigability either permanent or fluctuating with weakness of extra-ocular, facial, bulbar, axial, respiratory, or limb muscles. There is often hypotonia with developmental delay. Generalized muscle hypotonia and weakness, feeding difficulties, poor suck and cry, and developmental delay may also be the first signs of CMS [9].

Progressive respiratory failure is also associated with CMSs. In 2002, Byring et al. described sudden episodes of respiratory distress and bulbar weakness in CMSs elicited by infections, fever, and stress [10]. In some cases, significant apnea may require intubation and rapid initiation of ventilation support.

In our patient, the main signs of CMSs in the first month of life were chaotic movements followed by apnea needing ventilation without the typical clinical features of CMS.

The EEG did not show seizures and the observed chaotic movements of our patient have not been previously described in CMSs, in which muscular weakness and fatigue, with secondary clinical manifestations, are typical.

We believe that, during the first months of life of our patient, his muscular response was still not profoundly compromised so that chaotic movements mimicking seizures represented his physiological response to the reduced gas exchange due to respiratory impairment, triggered by different factors, such as viral infections.

In other words, before the typical respiratory insufficiency with apnea and the need for artificial ventilation of CMS-affected patients, he was still able during the early phase of his disease, he was still able to partially react to the developing asphyxia, i.e., acidosis and hypoxemia.

These chaotic movements soon followed by apnea with the need of assisted ventilation did not occur thereafter, i.e., after the age of six months. They could be interpreted as well as choking spells, already reported in patients with CMS, that occur when the musculature is not yet severely compromised [11].

CMSs are caused by mutations in genes that encode proteins involved in the organization, maintenance, function, or modification of the neuromuscular junction (NMJ) [3]. Age of onset, presenting symptoms, and response to treatment vary depending on the molecular mechanism that results from the underlying genetic defect. In our patient, we identified a mutation of the *COLQ* gene that encodes a multidomain functional protein of the NMJ crucial for anchoring acetyl cholinesterase (AChE) to the basal lamina and its accumulation at the NMJ [12].

Mutations in COLQ cause AChE deficiency. Patients severely affected by COLQ mutations abolish AChE activity present during infancy [3,13]. Less severely affected patients with residual enzyme activity present during childhood and become disabled later in life. The weakness can affect all voluntary muscles but can spare the ocular muscles, and in a few patients, the weakness has a limb–girdle distribution. Clinically, COLQ-related CMSs present with a broad range of features and severity, but the clinical manifestations are usually severe and can include respiratory failure, as in our patient [14].

However, biallelic COLQ mutations are responsible for a minority of CMSs cases with mutations that have been described in each of the three COLQ domains. To date, most reported COLQ mutations are uniformly distributed on the three conserved domains of COLQ protein: proline-rich attachment domain (PRAD) [exons 1–4] in the N-terminal region, heparan sulfate proteoglycan-binding domain (HSPBD) in the collagen domain [exons 4–14], and the C-terminal region [exons 15–17]. The causative mutations would exert their effects by different mechanisms resulting in prolonged synaptic currents and different action potentials due to expanded residence of acetylcholine in the synaptic space. Most of the *COLQ* gene mutations are nonsense, frameshift, splicing, or missense. Mutations localized in the N-terminal domain prevent the collagen domain from associating with the catalytic subunits, and those in the collagen domain affect the assembly of the triple-helical collagen domain. Most mutations in the C-terminal domain reduce COLQ expression or prevent the triple helical assembly (Figure 3A) [3].

To date, multiexon COLQ deletion has been described only in the two following cases. In 1998, Ohno et al. [15] reported a heterozygous truncation mutation consisting of a large-scale frameshift deletion [exon 2–3] in a CMS patient that abolished PRAD and followed the domain of the *COLQ* gene. Twenty years later, Wang et al. [16] identified a novel copy number deletion encompassing exon 14 and exon 15 of the *COLQ* gene in compound heterozygosity with the IVS16 + 3A > G variant.

Our case is the first report of a CMS harboring a homozygous extended deletion of 19.5 kb encompassing exon 11–17 of the *COLQ* gene (Figure 3A). This mutation truncates half of the collagen domain consisting of GXY triplets and one of the two HSPB domains as well as the entire C-terminal domain. It could be particularly damaging considering that the cationic residues of the HSPB domain interact with the anionic residues in the synaptic basal lamina and help the anchoring of the COLQ protein and the C-terminal region needed for assembly of the COLQ strands in a triple helix. The homozygous state of the deletion was identified. The crucial role of the domains involved could explain the severe clinical presentation in our patient. Congenital myasthenic syndromes are rare diseases; the prevalence of CMS is estimated at one tenth that of myasthenia gravis, which has a prevalence of 25:1.000.000–125:1.000.000 [11]. CMS also has genotypic and phenotypic heterogeneity; thus, therapy of CMSs is only symptomatic, and several drugs may exhibit severe side effects. The most frequently used drugs are AchE-inhibitors [14], 4-diaminopyridine [17], salbutamol [18], ephedrine [14], and fluoxetine [19].

In our case, the EMG revealed a myasthenic pattern. As our patient needed NIV during the night to sleep due to recurrent apnea, pyridostigmine was started. Only a partial response (reduction in palpebral ptosis) was obtained but after the molecular diagnosis, pyridostigmine was stopped and salbutamol was given, leading to a significant reduction in apnea and respiratory insufficiency. This finding highlights the importance of timely genetic counselling because the final diagnosis is crucial for correct CMSs treatment. Non-pharmacological treatments rely on physiotherapy, speech therapy, and occupational therapy. Sometimes invasive treatments are needed such as nasal intermittent positive pressure ventilation (NIPPV) during the night or the entire day in case of respiratory failure. PEG is used in the case of dysphagia, failure-to-thrive, or a nutritional disturbance. Surgical corrections are performed for severe deformities [20].

Prognosis and outcome of CMSs derive only from observational studies, case studies, and case reports because prospective outcome studies are not available. Due to the clinical variability, outcome and prognosis may vary considerably accordingly to various CMS types and infections. Fever or psychosocial stress could also have a negative effect on the outcome of these patients [1]. Our case report confirms the clinical heterogeneity of patients with CMS, indicating that recurrent apnea and respiratory crises during the first months of life could be responsible of electro-clinic seizures.

## Figures and Tables

**Figure 1 genes-11-01519-f001:**
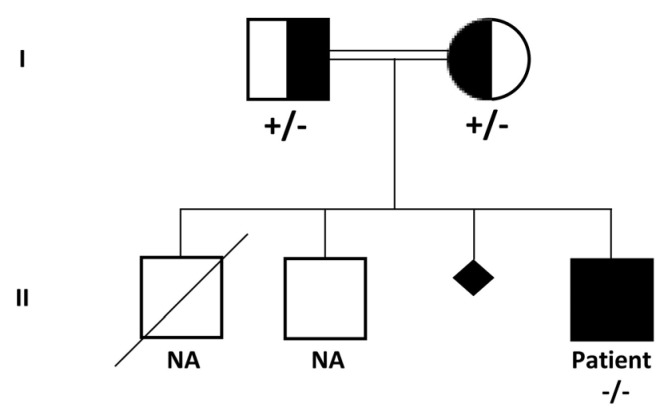
Pedigree chart of the family. Squares and circles indicate men and women, respectively. A diagonal line through a symbol indicates a deceased individual. A small rhombus indicates a miscarriage. Affected individuals are indicated by filled symbols. Carrier individuals are indicated by filled-empty symbols. II-4 genetic analysis revealed a homozygous microdeletion involving the 3p25.1 chromosome region that contains part of the collagenic tail of endplate acetylcholinesterase (COLQ) gene. The microdeletion resulted in heterozygous status in I-1 and I-2.

**Figure 2 genes-11-01519-f002:**
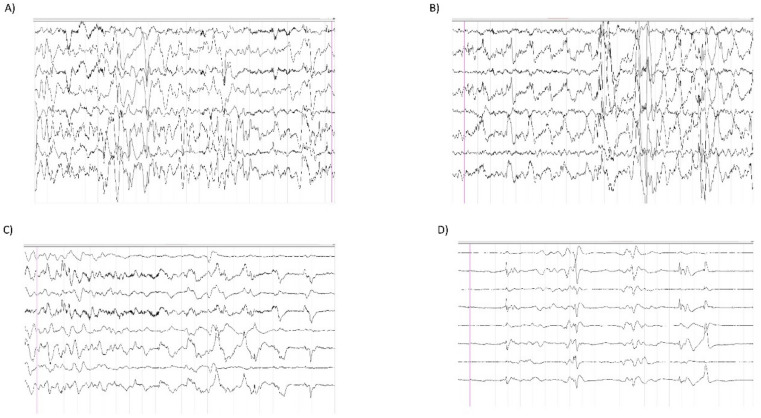
Electroencephalography (EEG). (**A**,**B**) High voltage slow waves from right occipital areas with bilateral involvement. Video EEG recording showed associated chaotic movements. (**C**,**D**) Occipital bilateral slow waves with suppression of electrical activity. Video EEG recording showed apnea and lack of motion.

**Figure 3 genes-11-01519-f003:**
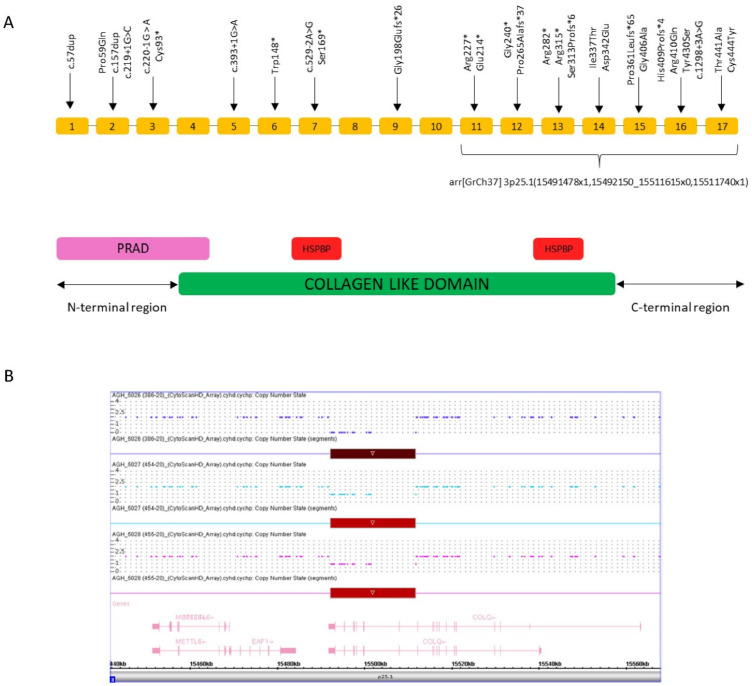
Schematic representation of the *COLQ* gene and results of single nucleotide polymorphism (SNP)-array analysis in the patient and his parents (**A**) COLQ exons with 28 published pathogenic variants (upper part) and the microdeletion described in this study (lower part). Three COLQ domains: (1) conserved domains of COLQ include an N-terminal proline-rich attachment domain (PRAD) that associates each COLQ strand with an acetylcholinesterase tetramer, (2) a central collagen domain that contains two heparan sulfate proteoglycan binding (HSPBP) domains, and (3) a C-terminal region needed for assembly of the COLQ strands in a triple helix. (**B**) Results of SNP-array analysis in the patient and his parents. The copy number state of each probe is drawn along chromosome 3 from 15.44 to 15.56 Mb (University of California Santa Cruz (UCSC) Genome Browser, buildGRCh37/hg19). The upper panel represents the copy number state of the proband, the middle panel that of the father, and the lower panel that of the mother. Values on the *Y*-axis indicate the inferred copy number according the probes’ intensities. Red bars indicate the deletion identified in the patient (homozygous state, copy number = 0) and his parents (heterozygous state, copy number = 1).

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
