# Peer review of "The First Case of Congenital Myasthenic Syndrome Caused by a Large Homozygous Deletion in the C-Terminal Region of COLQ (Collagen Like Tail Subunit of Asymmetric Acetylcholinesterase) Protein"

_genes, 2020, doi:10.3390/genes11121519_

Round 1
Reviewer 1 Report
A very interesting work, with a good clínical and genetic description
Author Response
We wish to thank very much you for your careful and favourable review of the manuscript.
Best Regards,
Nicoletta Resta
Please see the attachment:
English Editing Certificate

Reviewer 2 Report
The authors report the phenotype of one young child suffering from epileptic seizure in his neonatal period then fatigable muscle weakness, the latter suggesting congenital myasthenic syndrome (CMS). They performed clinical exome sequencing and CNV analyses and identified in this patient one large and inherited homozygous deletion encompassing the last 7 exons of the COLQ gene known to cause CMS with acetylcholinesterase deficiency. This gene is well known to cause CMS with several patients and mutations (mostly point mutations), already described. To my opinion, the main interests of the manuscript lie on the observation of epileptic seizures at birth in this patient with CMS, and the realization of both clinical exome sequencing and CNV analyses for medical purpose on the same sample to detect the deletion. However, the manuscript does not emphasize a lot on these two points and could be improved to achieve this goal. I therefore suggest the following changes.
- please remind the most frequent inheritance pattern of CMS (recessive) somewhere in the introduction
- I don’t understand exactly what the authors mean by “specific clinical syndromic phenotype” (line 52)
- Indicates that abnormalities of EMG specific to CMS are decrement at repetitive nerve stimulation (RNS) frequency (usually at 3Hz). As doing such a physiological analysis in a young child is challenging, could the authors precise what was the EMG myastenic pattern (line 135 : what was the nerve-muscle couple analyzed, what was the frequency of repetitive nerve stimulation used,) and their findings (decrement rate for example, and whether a post-increment was also observed)? Was single fiber EMG also performed? May be the authors could put the EMG results in figure 2 with EEG as they are important for the diagnosis of CMS? Was some muscle biopsy available to test AChE deficiency by histochemical staining for diagnosis purpose?
- The authors performed both clinical exome and SNP arrays analyses but they only refer to their SNP arrays results in the dedicated section. Were other candidate variants identified by exome sequencing, especially in relation with the neonatal epileptic phenotype? In the method section, the authors indicate that they used SOPHiA DDM to analyze CNV on exome sequencing data. Could they indicate what were the results for the COLQ gene, especially for the deleted exons?
- Discussion could be improved by best comparing their report with previous reports of patients with CMS due to AChE deficiency. Could the authors best indicate how many patients with CMS due to AChE deficiency are described in the literature (line 203)? If I’m right, neonate epileptic seizures are more frequent than CMS. Therefore, the association with CMS due to AChE deficiency may be due to chance so I would not be so affirmative in the conclusion (see final sentence lines 230-232). If their patient is the only one with associated epilepsy amongst several patients already described, this argues more for an association due to chance. On the other side, I was told that episodes of respiratory distress in neonates are a main feature in severe forms of CMS (so please modify accordingly the sentences lines 171-173) and may be mistaken for epileptic seizures by pediatricians not aware of CMS. This could be best discussed.
- Pyridostigmine is an AChE blocker and is contraindicated in CMS with AChE. Could the authors further underline this important fact both in the results and discussion sections as it underlines the importance of doing genetic counselling in a short delay? What was the pyridostigmine dosage for the patient and how did he react to it?
Minor points
- Lines 56-60 are not wrong but not really exact at the biological level…The best would be maybe to refer to the genes, for example : “The most common cause of CMS are molecular defects impacting the function (CHRNE..) or the clustering (RAPSN, DOK7, MUSK, LRP4) of muscle nicotinic acetylcholine receptors at the neuromuscular junction. CMS can also be caused by mutations in genes encoding proteins in the synaptic basement membrane (COLQ, AGRN,…), in the presynaptic release machinery (CHAT, VACHT, SLC5A7, SYT2…) or in protein glyscosylation (…)...”
- COLQ in italics when referring to the gene, ColQ when referring to the protein Please reformulate the role of ColQ, for example as follows: “ColQ forms a triple helix collagenic tail required for the anchoring of acetylcholinesterase to the synaptic basement membrane”.
- Line 68: The C-terminal region of the ColQ protein (or the deletion of the last exons of the COLQ gene)
- I recommend to speak about “precision medicine” rather than “personalized pharmacologic treatment”
- In Materials and methods section, could the authors indicate the characteristics (how many bases genes sequenced, mean coverage of bases) for the SOPHiA panel kit? Also indicate how many SNP are analyzed with the High Resolution SNP-Array analysis.
- Figure 1 : the neonatal death of the third infant may suggest he/she suffered from the same disease than the proband. Was the cause of the neonatal death really unknown ? I’m not sure that this dead infant could be indicated as “uneventful pregnancy” in Figure 1 legend.
- Figure 3 : A and B have to be replaced each other to fit with the legend
- Line 212, prevalence : Do the authors mean “25:1.000.000-125:1.000.000” for myasthenia gravis prevalence (and what is the reference for this rate?)
Author Response
Re: revision of genes-1023541
Type of manuscript: Case Report
Title: The First case of Congenital Myasthenic Syndrome caused by a large
homozygous deletion in the C-terminal region of COLQ (Collagen Like Tail
Subunit of Asymmetric Acetylcholinesterase) gene
Please find attached and uploaded to your website the revised version (R1) of the above manuscript.
We wish to thank very much you and both your expert reviewers for their careful review of the above manuscript. We revised the manuscript based on the comments provided by reviewer 2 and hopefully, the present revised manuscripts meet the criteria for publication.
Please find here below our response to the comments.
Reviewer #2
Comments to the Author
Reviewer 2
The authors report the phenotype of one young child suffering from epileptic seizure in his neonatal period then fatigable muscle weakness, the latter suggesting congenital myasthenic syndrome (CMS). They performed clinical exome sequencing and CNV analyses and identified in this patient one large and inherited homozygous deletion encompassing the last 7 exons of the COLQ gene known to cause CMS with acetylcholinesterase deficiency. This gene is well known to cause CMS with several patients and mutations (mostly point mutations), already described. To my opinion, the main interests of the manuscript lie on the observation of epileptic seizures at birth in this patient with CMS, and the realization of both clinical exome sequencing and CNV analyses for medical purpose on the same sample to detect the deletion. However, the manuscript does not emphasize a lot on these two points and could be improved to achieve this goal. I therefore suggest the following changes.
- please remind the most frequent inheritance pattern of CMS (recessive) somewhere in the introduction
- As suggested, we have included this information. [lines 44-45]
- I don’t understand exactly what the authors mean by “specific clinical syndromic phenotype” (line 52)
- We deleted this sentence and added some clinical features to better define the related phenotype. [line 53]
- Indicates that abnormalities of EMG specific to CMS are decrement at repetitive nerve stimulation (RNS) frequency (usually at 3Hz). As doing such a physiological analysis in a young child is challenging, could the authors precise what was the EMG myastenic pattern (line 135 : what was the nerve-muscle couple analyzed, what was the frequency of repetitive nerve stimulation used,) and their findings (decrement rate for example, and whether a post-increment was also observed)? Was single fiber EMG also performed?
- We corrected the text, which now reads, “The EMG revealed a decremental response to repetitive nerve stimulations on the deltoid muscle at a frequency of 3 Hz. It showed a decrease in the amplitude of the motor unit potential (MUP) with no post-increment. [lines 139-141]
A single-fiber exam was not performed.”
- May be the authors could put the EMG results in figure 2 with EEG as they are important for the diagnosis of CMS?
- The results were available as a final report only. Photos of the EMG were not available.
- Was some muscle biopsy available to test AChE deficiency by histochemical staining for diagnosis purpose?
- The parents disallowed a muscle biopsy.
- The authors performed both clinical exome and SNP arrays analyses but they only refer to their SNP arrays results in the dedicated section. Were other candidate variants identified by exome sequencing, especially in relation with the neonatal epileptic phenotype? In the method section, the authors indicate that they used SOPHiA DDM to analyze CNV on exome sequencing data. Could they indicate what were the results for the COLQ gene, especially for the deleted exons?
- The detailed information obtained via CES has been added as requested. [lines 151-156]
- Discussion could be improved by best comparing their report with previous reports of patients with CMS due to AChE deficiency. Could the authors best indicate how many patients with CMS due to AChE deficiency are described in the literature (line 203)? If I’m right, neonate epileptic seizures are more frequent than CMS. Therefore, the association with CMS due to AChE deficiency may be due to chance so I would not be so affirmative in the conclusion (see final sentence lines 230-232). If their patient is the only one with associated epilepsy amongst several patients already described, this argues more for an association due to chance. On the other side, I was told that episodes of respiratory distress in neonates are a main feature in severe forms of CMS (so please modify accordingly the sentences lines 171-173) and may be mistaken for epilepticseizures by pediatricians not aware of CMS. This could be best discussed.
- The Discussion was improved by adding the details suggested. [lines 183-186,197-198, 221-222]
- Pyridostigmine is an AChE blocker and is contraindicated in CMS with AChE. Could the authors further underline this important fact both in the results and discussion sections as it underlines the importance of doing genetic counselling in a short delay? What was the pyridostigmine dosage for the patient and how did he react to it?
- These details were added as suggested. [lines 234-239]
Minor points
- Lines 56-60 are not wrong but not really exact at the biological level…The best would be maybe to refer to the genes, for example : “The most common cause of CMS are molecular defects impacting the function (CHRNE..) or the clustering (RAPSN, DOK7, MUSK, LRP4) of muscle nicotinic acetylcholine receptors at the neuromuscular junction. CMS can also be caused by mutations in genes encoding proteins in the synaptic basement membrane (COLQ, AGRN,…), in the presynaptic release machinery (CHAT, VACHT, SLC5A7, SYT2…) or in protein glyscosylation (…)...”
- We agree and changed the sentence as suggested. [lines 60-65]
- COLQ in italics when referring to the gene, ColQ when referring to the protein Please reformulate the role of ColQ, for example as follows: “ColQ forms a triple helix collagenic tail required for the anchoring of acetylcholinesterase to the synaptic basement membrane”.
- These changes have been made. [line 69-71]
- Line 68: The C-terminal region of the ColQ protein (or the deletion of the last exons of the COLQ gene)
- This change was made. [line 73-74]
- I recommend to speak about “precision medicine” rather than “personalized pharmacologic treatment”
- This change was made. [line 77]
- In Materials and methods section, could the authors indicate the characteristics (how many bases genes sequenced, mean coverage of bases) for the SOPHiA panel kit? Also indicate how many SNP are analyzed with the High Resolution SNP-Array analysis.
- These details have been added. [lines101-102]
- Figure 1 : the neonatal death of the third infant may suggest he/she suffered from the same disease than the proband. Was the cause of the neonatal death really unknown ? I’m not sure that this dead infant could be indicated as “uneventful pregnancy” in Figure 1 legend.
- The modification suggested has been made. [line 116] Unfortunately, the cause of the neonatal death is unknown; it occurred at the home (in Pakistan).
- Figure 3 : A and B have to be replaced each other to fit with the legend
- The title was modified, and the error was corrected in the legend. [line 164-170]
- Line 212, prevalence : Do the authors mean “25:1.000.000-125:1.000.000” for myasthenia gravis prevalence (and what is the reference for this rate?) NL
- This error was corrected, and a citation was added. [line 230]
Round 2
Reviewer 2 Report
I thank the authors for their answers and prompt reply. However, I'm still a little bit dissapointed by their clinical discussion regarding the relationship between apparent seizures (EEG pattern with video), and CMS in their patient, and to my opinion this should be really improved for readers not familiar with severe forms of CMS. Title (C-terminal deletion in the COLQ gene....) and abstract have not been modified in response to my previous remarks on gene/protein (especially this sentence for the latter "Among these, the COLQ gene (Collagenic Tail of Endplate Acetylcholinesterase; OMIM25 603033) has a crucial role in anchoring the enzyme into the synaptic basal lamina. Here we report
on the first case of a patient with a homozygous deletion affecting the C-terminal of the COLQ gene in a CMS patient born to consanguineous parents of Pakistani origin.")
Please precise the decrement value for EMG.
The extent of the COLQ gene deletion identified by CES (exons 13-17) and SNP (exons 11-17) analyses is not equal. Could the authors indicate how many kb this difference represent and discuss its possible origin (for example may be it due to a low number of SNPs in exons 11 and 12 used for CNV analyses)?
The authors could indicate that the COLQ deletion identifed by Wang et al was heterozygous, associated to a splice variant on the other allele.
Author Response
Editor-Genes
Re: revision of genes-1023541
Type of manuscript: Case Report
Title: The First case of Congenital Myasthenic Syndrome caused by a large
homozygous deletion in the C-terminal region of COLQ (Collagen Like Tail
Subunit of Asymmetric Acetylcholinesterase) gene
Please find attached and uploaded to your website the revised version (R2) of the above manuscript.
Please find here below our response to the comments.
Reviewer #2
- I thank the authors for their answers and prompt reply. However, I'm still a little bit dissapointed by their clinical discussion regarding the relationship between apparent seizures (EEG pattern with video), and CMS in their patient, and to my opinion this should be really improved for readers not familiar with severe forms of CMS.
- Thank you for your comments. We agree with Reviewer #2 that our previous submission was not entirely clear and acceptable.
- We have described better the EEG results avoiding the therm “paroxysmal acitivity”, because the pattern was characterized by uniform high-voltage waves that could not be classified as electrical seizures, according to 2017 ILAE Newborn seizures guidelines.
- The clinical apparent seizures of our patient, but indeed chaotic movements, could be the response to his impending respiratory insufficiency in the early phase of this type of CMS, when muscular weakness is still mild. The developing acidosis and hypoxemia were still able to stimulate his respiratory effort. No description of similar neonatal signs of this type of CMS have been found, but choking spells. [line 175,186-191],
- Title (C-terminal deletion in the COLQ gene....) and abstract have not been modified in response to my previous remarks on gene/protein (especially this sentence for the latter "Among these, the COLQ gene (Collagenic Tail of Endplate Acetylcholinesterase; OMIM25 603033) has a crucial role in anchoring the enzyme into the synaptic basal lamina. Here we report
on the first case of a patient with a homozygous deletion affecting the C-terminal of the COLQ gene in a CMS patient born to consanguineous parents of Pakistani origin.")
- These changes were made. [line 4-5; 24-26]
- Please precise the decrement value for EMG.
- EMG's final report is attached to this file, it doesn’t report the decrement value.
- The extent of the COLQ gene deletion identified by CES (exons 13-17) and SNP (exons 11-17) analyses is not equal. Could the authors indicate how many kb this difference represent and discuss its possible origin (for example may be it due to a low number of SNPs in exons 11 and 12 used for CNV analyses)?
- The CES results were improved by adding the details suggested [line 203-205].
- The authors could indicate that the COLQ deletion identifed by Wang et al was heterozygous, associated to a splice variant on the other allele.
- This detail has been added. [line 224]
